# Improved Medium Baseline RTK Positioning Performance Based on BDS/Galileo/GPS Triple-Frequency-Only Observations

Xifeng Dang [1], Xiao Yin [1,2,3], Yize Zhang [4], Chengfa Gao [3,*], Jincheng Wu [1] and Yongqiang Liu [1]

1   China Railway First Group Urban Rail Transit Engineering Co., Ltd., Wuxi 214000, China;
    cgdangxifeng@crecg.com (X.D.); yinxiaotongji@zafu.edu.cn (X.Y.); wujincheng@crecg.com (J.W.);
    liuyongqiang820@crecg.com (Y.L.)
2   College of Environment and Resource Science, Zhejiang A&F University, Hangzhou 311300, China
3   School of Transportation, Southeast University, Nanjing 210096, China
4   Shanghai Astronomical Observatory, Chinese Academy of Sciences, Shanghai 200030, China;
    zhyize@shao.ac.cn
*   Correspondence: gaochfa@seu.edu.cn

**Abstract:** With the global service of the BeiDou Navigation Satellite System (BDS), the Galileo Navigation Satellite System (Galileo), and the modernization of the Global Positioning System (GPS), achieving high-precision positioning through triple-frequency-only observations in medium baseline real-time kinematics (RTK) is anticipated. This study investigates the impacts of double-difference (DD) troposphere delay and ionosphere delay on ambiguity resolution (AR) based on six medium baselines at a latitude of 30°. Additionally, it evaluates positioning accuracy, fixing rate, convergence time, and computational time using triple-frequency-only (B1I/B2a/B3I, E1/E5a/E5b, L1/L2/L5) data, comparing these results to those obtained from dual-frequency (B1I/B2a, E1/E5a, L1/L2) and combined dual-frequency and triple-frequency data. The experimental findings suggest that, for geometry-based wide-lane (WL) AR, the DD troposphere delay and ionosphere delay can be disregarded. However, they cannot be overlooked when aiming to resolve the raw ambiguity. Triple-frequency-only RTK exhibits comparable positioning accuracy to dual-frequency RTK, with its primary advantage lying in faster convergence. The probability of achieving convergence within 180 s is approximately 8.0% higher for triple-frequency-only RTK compared to dual-frequency RTK. In terms of computational time, the use of triple-frequency-only data reduces the required time by 8.26 s compared to the approach that simultaneously employs both dual-frequency and triple-frequency data, resulting in a computational time reduction of approximately 20%. Therefore, when conducting medium baseline RTK positioning, it is recommended to adopt the ambiguity resolution method proposed in this paper based on triple-frequency-only observations.

**Keywords:** BDS/Galileo/GPS; medium baseline RTK; triple-frequency-only; ambiguity resolution; double-difference atmosphere delay; convergence; computation cost time

## 1. Introduction

At present, satellite positioning has been widely used in people's daily life and has become an indispensable location service technology. In addition to traditional Global Navigation Satellite Systems (GNSS), i.e., GPS and GLONASS, Galileo and BeiDou Navigation Satellite System (BDS) are fully operational and available. Currently, GPS adds L5 frequency after BLOCK IIF satellites, so that about half of GPS satellites can transmit triple-frequency (L1/L2/L5) signals [1]. Galileo and BDS can transmit five-frequency signals, i.e., E1/E5a/E5b/E5/E6 for Galileo and B1C/B1I/B2a/B2b/B3I for BDS3 (the third phase of BDS) [2]. Multi-GNSS and multi-frequency signals can improve the satellite geometric configuration and shorten the convergence time [3,4], thereby further expanding the application of satellite positioning.

Based on double-difference (DD) carrier phase residuals of zero baselines, Quan et al. [5] found that the phase observation accuracy for L1/L2/E5a/E5b is between 1.0 and 2.0 mm, while that for B1I is over 2 mm when DD is performed with two satellites from two different orbit types. Xie et al. [6] showed that B2a/B2b/L5/E5a/E5b have similar signal strength by analyzing the carrier-to-noise density ratio ($C/N_0$). They also expressed that code observation noise of B1C/B2a/B2b have no significant difference compared to those of L1/L5 and E1/E5a/E5b. In an in-depth assessment of new observation signals (i.e., B1C and B2a) from BDS-3, Zhang et al. [7] concluded that the phase observation accuracy of B1C/B2a is better than that of B1I/B2I/B3I and BDS-3/GPS/Galileo medium earth orbit (MEO) satellites have the same level of signal strength, code, and phase observation accuracy at the interoperable frequency. These research results are very useful for the establishment of multi-GNSS stochastic model.

With code and phase observations, real-time kinematic (RTK) technology is well-known in achieving millimeter- to centimeter-level positioning. For short baselines (<10 km), DD between two observing stations and two satellites can eliminate satellite orbit errors, satellite and receiver clock offset/hardware bias, and atmospheric delay, thus ensuring the integer characteristic of ambiguity [8,9]. Odijk et al. [10] first evaluated GPS/Galileo single-frequency short baseline RTK and found that, when compared to GPS-only strategy, GPS/Galileo ambiguity resolution (AR) is significantly improved. Odolinski et al. [11] combined single-frequency short baseline observations of GPS, Galileo, BDS, and Quasi-Zenith Satellite System (QZSS) and showed a significant improvement of the time-to-correct-fix. Using dual-frequency observations of GPS and BDS, Brack [12] also confirmed the benefit of combining two GNSS for short baselines. Wu et al. [13] assessed GPS/BDS/Galileo dual-frequency short baseline RTK and obtained lower ambiguity dilution of precision (ADOP) and higher ratios.

For medium to long baselines, the DD method finds it difficult to eliminate ionosphere and troposphere delay errors, which hinders rapid AR. Feng [14] utilized ionosphere-free (IF) combination to eliminate ionosphere delay errors; however, this could not be beneficial for signal reacquisition and rapid re-convergence after signal loss. Thus, raw code and phase measurements are adopted in [15], with DD ionosphere delay being estimated as a parameter. By using triple-frequency observations, Gao et al. [16] improved GPS/BDS AR for medium baselines. Zhang et al. [17] presented a detailed description of the AR strategy for extra-wide-lane (EWL), wide-lane (WL), and raw N1/N2 ambiguity, and analyzed contribution of QZSS to multi-GNSS long baseline RTK. In [16,17], non-triple-frequency satellite observations were also used due to the limited availability of triple-frequency satellites. However, with a substantial number of satellites now capable of transmitting triple-frequency data, it is necessary to examine the performance of medium baseline RTK using triple-frequency-only satellite data, which is very important for high-frequency on-embedded terminals, such as 10 Hz, as they have limited computing power and significant latency.

Up to now, there are approximately 16 GPS satellites capable of transmitting triple-frequency signals (L1/L2/L5), about 24 Galileo satellites capable of transmitting triple-frequency signals (E1/E5a/E5b), and about 26 BDS satellites capable of transmitting triple-frequency signals (B1I/B2a/B3I). Therefore, in this research, we mainly present the medium baseline RTK positioning performance based on triple-frequency-only satellite data of BDS/Galileo/GPS. The algorithm for medium baseline triple-frequency RTK is presented in detail. Then, based on the analysis of DD ionosphere delay and troposphere delay using real baseline data, the positioning performance of dual-frequency, dual-frequency/triple-frequency, and triple-frequency-only RTK is provided, including positioning accuracy, fixing rate, convergence time, and computational cost time. Finally, the results are summarized in the conclusion.

## 2. Medium Baseline RTK Algorithm

### 2.1. Double-Difference Triple-Frequency Measurement Model

Considering DD ionosphere delay and troposphere delay residuals as the parameters to be estimated, the original triple-frequency DD code and phase observation equations are given [18] as

$$
\begin{cases}
\nabla\Delta P_1 = \nabla\Delta\rho_{rb}^{st} + m^{st}\nabla\Delta T_{rb} + \alpha_1\nabla\Delta I_{rb}^{st} + e_{rb,1}^{st} \\
\nabla\Delta P_2 = \nabla\Delta\rho_{rb}^{st} + m^{st}\nabla\Delta T_{rb} + \alpha_2\nabla\Delta I_{rb}^{st} + e_{rb,2}^{st} \\
\nabla\Delta P_3 = \nabla\Delta\rho_{rb}^{st} + m^{st}\nabla\Delta T_{rb} + \alpha_3\nabla\Delta I_{rb}^{st} + e_{rb,3}^{st} \\
\nabla\Delta L_1 = \nabla\Delta\rho_{rb}^{st} + m^{st}\nabla\Delta T_{rb} - \alpha_1\nabla\Delta I_{rb}^{st} + \lambda_1\nabla\Delta N_1^{st} + \varepsilon_{rb,1}^{st} \\
\nabla\Delta L_2 = \nabla\Delta\rho_{rb}^{st} + m^{st}\nabla\Delta T_{rb} - \alpha_2\nabla\Delta I_{rb}^{st} + \lambda_2\nabla\Delta N_2^{st} + \varepsilon_{rb,2}^{st} \\
\nabla\Delta L_3 = \nabla\Delta\rho_{rb}^{st} + m^{st}\nabla\Delta T_{rb} - \alpha_3\nabla\Delta I_{rb}^{st} + \lambda_3\nabla\Delta N_3^{st} + \varepsilon_{rb,3}^{st}
\end{cases}
\tag{1}
$$

where $\nabla\Delta$ is the DD operator; $P_i$ and $L_i$ denote the code and phase observations, respectively, with $i = 1, 2, 3$ being the frequency; $\rho_{rb}^{st}$ is the geometry distance between the receiver and satellite, with $b, r$ being the base station and the rover station and $t, s$ being the pivot satellite and the rover satellite; $T$ denotes the zenith tropospheric delay with the mapping function $m$; $I$ represents the slant ionospheric delay with the coefficient $\alpha_i = \frac{\lambda_i^2}{\lambda_1^2}$ depending on the wavelength $\lambda_i$; $N_i$ refers to the phase ambiguity; and $e$ and $\varepsilon$ denote the code and phase observation noise. In this research, for GPS, the three frequencies refer to L1/L2/L5; for Galileo, the three frequencies refer to E1/E5a/E5b; for BDS, the three frequencies refer to B1I/B2a/ B3I.

Based on the raw code and phase observation equations, the combined DD code and phase are defined as

$$
\begin{aligned}
\nabla\Delta P_{(i,j,k)} &= \frac{i \cdot f_1 \cdot \nabla\Delta P_1 + j \cdot f_2 \cdot \nabla\Delta P_2 + k \cdot f_3 \cdot \nabla\Delta P_3}{i \cdot f_1 + j \cdot f_2 + k \cdot f_3} \\
\nabla\Delta L_{(i,j,k)} &= \frac{i \cdot f_1 \cdot \nabla\Delta L_1 + j \cdot f_2 \cdot \nabla\Delta L_2 + k \cdot f_3 \cdot \nabla\Delta L_3}{i \cdot f_1 + j \cdot f_2 + k \cdot f_3}
\end{aligned}
\tag{2}
$$

where the coefficients $i, j, k$ are arbitrary integers and $f_i = \frac{c}{\lambda_i}$ denotes the frequency, with $c$ being the speed of light. The combined wavelength $\lambda_{(i,j,k)}$ is defined as

$$
\lambda_{(i,j,k)} = \frac{c}{i \cdot f_1 + j \cdot f_2 + k \cdot f_3}
\tag{3}
$$

and the combined ambiguity is

$$
N_{(i,j,k)} = i \cdot \nabla\Delta N_1 + j \cdot \nabla\Delta N_2 + k \cdot \nabla\Delta N_3
\tag{4}
$$

The combined ionosphere delay coefficient is

$$
\alpha_{(i,j,k)} = \frac{f_1^2\left(\frac{i}{f_1} + \frac{j}{f_2} + \frac{k}{f_3}\right)}{i \cdot f_1 + j \cdot f_2 + k \cdot f_3}
\tag{5}
$$

For GPS, when $(i, j, k) = (0, 1, -1)$ and $(i, j, k) = (1, -1, 0)$, the wavelengths $\lambda_{(0,1,-1)}$ and $\lambda_{(1,-1,0)}$ are about 5.86 m and 0.86 m, typically referred to as EWL and WL, showing that their ambiguities are easy to be fixed.

### 2.2. Triple-Frequency Ambiguity Resolution

As shown in Equation (1), the raw ambiguity parameter is related to the slant ionosphere delay parameter and requires a long time to be resolved. To address this, the ambiguity of EWL and WL is first fixed through combined observations of EWL and WL.

Using geometry-free and ionosphere-free (GFIF) combination, the float EWL ambiguity can be

$$\nabla\Delta\hat{N}_{(0,1,-1)} = \frac{\nabla\Delta L_{(0,1,-1)} - \nabla\Delta P_{(0,1,1)}}{\lambda_{(0,1,-1)}} \tag{6}$$

where $\nabla\Delta\hat{N}_{(0,1,-1)}$ is the float EWL ambiguity. Due to the long wavelength, the float EWL ambiguity is fixed by rounding to the nearest integer. After the EWL ambiguity fixed, the EWL phase observation equation can be given as

$$\nabla\Delta L_{(0,1,-1)} - \lambda_{(0,1,-1)}\nabla\Delta N_{(0,1,-1)} = \nabla\Delta\rho_{rb}^{st} + \varepsilon_{rb,(0,1,-1)}^{st} \tag{7}$$

As shown in Equation (7), the ambiguity-fixed EWL phase observation can be regarded as a high-precision code observation, thus accelerating the WL AR. This is also the main reason why the triple-frequency RTK converges faster than the traditional dual-frequency RTK. Here, the DD ionosphere delay and troposphere delay residuals are ignored due to their small magnitude [19,20].

Due to the shorter wavelength compared to the EWL ambiguity, it is prone to wrongly fix when directly rounding WL ambiguity. Thus, the geometry-based approach is adopted:

$$\begin{cases} \nabla\Delta L_{(1,-1,0)} = \nabla\Delta\rho_{rb}^{st} + \lambda_{(1,-1,0)}\nabla\Delta\hat{N}_{(1,-1,0)} + \varepsilon_{rb,(1,-1,0)}^{st} \\ \nabla\Delta L_{(0,1,-1)} - \lambda_{(0,1,-1)}\nabla\Delta N_{(0,1,-1)} = \nabla\Delta\rho_{rb}^{st} + \varepsilon_{rb,(0,1,-1)}^{st} \end{cases} \tag{8}$$

As shown in Equation (8), the ambiguity-fixed EWL phase observation is used for fast WL AR. It is worth noting that, for traditional dual-frequency RTK, in Equation (8), there are no high-precision ambiguity-fixed EWL phase observations available, which are replaced by $\nabla\Delta P_1$ and $\nabla\Delta P_2$, as shown in [17], resulting in slow convergence. Also, the DD ionosphere delay and troposphere delay residuals are ignored; the rationale for this is explained in the next section. When obtaining the float WL ambiguity $\nabla\Delta\hat{N}_{(1,-1,0)}$, the least-square ambiguity decorrelation adjustment (LAMBDA) is used for AR [21]. After resolving WL ambiguity, the ambiguity-fixed WL phase observation can be regarded as a code observation again, but with significantly reduced noise compared to the raw code observation. Then, the float N1/N2/N3 ambiguities can be solved as follows:

$$\begin{bmatrix} V_{\nabla\Delta P_1} \\ V_{\nabla\Delta P_2} \\ V_{\nabla\Delta P_3} \\ V_{\nabla\Delta L_1} \\ V_{\nabla\Delta L_2} \\ V_{\nabla\Delta L_3} \\ V_{\nabla\Delta L_{(1,-1,0)}} \end{bmatrix} = \begin{bmatrix} e & m^{st} & \alpha_1 & 0 & 0 & 0 \\ e & m^{st} & \alpha_2 & 0 & 0 & 0 \\ e & m^{st} & \alpha_3 & 0 & 0 & 0 \\ e & m^{st} & -\alpha_1 & 1 & 0 & 0 \\ e & m^{st} & -\alpha_2 & 0 & 1 & 0 \\ e & m^{st} & -\alpha_3 & 0 & 0 & 1 \\ e & m^{st} & \frac{f_1}{f_2} & 0 & 0 & 0 \end{bmatrix} \begin{bmatrix} X \\ \nabla\Delta T_{rb} \\ \nabla\Delta I_{rb}^{st} \\ \lambda_1\nabla\Delta\hat{N}_1^{st} \\ \lambda_2\nabla\Delta\hat{N}_2^{st} \\ \lambda_3\nabla\Delta\hat{N}_3^{st} \end{bmatrix} - \begin{bmatrix} \nabla\Delta P_1 - \nabla\Delta P_1^0 \\ \nabla\Delta P_2 - \nabla\Delta P_2^0 \\ \nabla\Delta P_3 - \nabla\Delta P_3^0 \\ \nabla\Delta L_1 - \nabla\Delta L_1^0 \\ \nabla\Delta L_2 - \nabla\Delta L_2^0 \\ \nabla\Delta L_3 - \nabla\Delta L_3^0 \\ \nabla\Delta L_{(1,-1,0)} - \lambda_{(1,-1,0)}\nabla\Delta N_{(1,-1,0)} - \nabla\Delta L_{(1,-1,0)}^0 \end{bmatrix} \tag{9}$$

where $V$ refers to the code or phase noise; $e$ is the geometry vector; $X$ is the baseline coordinate vector; and $\nabla\Delta P_i^0$ and $\nabla\Delta L_i^0$ denote the computed code and phase observation, respectively. Therefore, the last column of Equation (9) represents the OMC (observed minus computed) vector. Due to the fixed WL ambiguity $\nabla\Delta N_{(1,-1,0)}$, the precision of float N1/N2/N3 ambiguities is improved, thus being beneficial for fast AR.

It is worth noting that, during the algorithm program implementation, two threads are employed. One thread fixes the WL ambiguity $\nabla\Delta\hat{N}_{(1,-1,0)}$ according to Equations (6)–(8), while another thread estimates the raw float N1/N2/N3 ambiguities using Equation (1). If the first thread successfully fixes the WL ambiguity, the ambiguity-fixed phase observations are utilized as virtual observations for the second thread's filter updating process. In case the WL ambiguity remains unfixed in the first thread, the algorithm proceeds to the next epoch.

*2.3. Determination Process Noise of DD Ionospheric and Tropospheric Delay*

The model strength is limited because Equation (9) considers DD ionosphere delay and troposphere delay as unknown parameters. To improve the model strength, we estimate DD ionosphere delay and troposphere delay as random walk parameters in Kalman filter [15].

The DD zenith troposphere delay is correlated with baseline distance and station height, and its random walk can be described as follows [17]:

$$Q_{\nabla\Delta T} = \left( log\left(1 + D \cdot 10^{-5}\right) \cdot 0.02 + H \cdot 10^{-5} \right)/\sqrt{3600 \cdot dt} \tag{10}$$

where *D* is the baseline distance; *H* is the height difference, measured in meters; and *dt* is the epoch time difference, measured in seconds.

Since the estimated ionosphere delay is the slant, it is important to consider not only the baseline length and latitude dependence, but also its correlation with elevation angle. Therefore, the random walk noise of DD ionosphere delay can be given as follows [17]:

$$Q_{\nabla\Delta I} = \left( D \cdot 5 \cdot 10^{-6} \cdot exp((90 - lat)/50 - 1) \right)/sin(El)\sqrt{3600 \cdot dt} \tag{11}$$

where *lat* refers to the average latitude of the base station and the rover station and *El* is the elevation angle of the satellite.

In addition, we set the priori DD ionosphere delay and troposphere delay to zero, and give them a large initial standard deviation, i.e., 1.5 m and 0.2 m, respectively, thus avoiding introducing bias [16]. If these initial standard deviation values are too small, the model strength can be enhanced, which may, however, result in a biased float solution when the ionosphere delay is active.

## 3. Datasets and Processing Strategies

To evaluate the performance of the proposed medium baseline RTK, we selected six baselines at latitude 30°, namely HZDQ-HZLA, HZDQ-HZXS, HZXS-HZLA, WHJA-EZEC, EZEC-WHJX, and WHJX-WHJA, with the first three baselines located in Hangzhou and the remaining three baselines located in Wuhan. The distances of these baselines range from 45 km to 66 km, with data collection taking place on 23 May 2022, which corresponds to the DOY (day of year) 143. The distribution of these stations can be seen in Figure 1. These stations are equipped with a Septentrio PolaRx4 receiver, receiving BDS/Galileo/GPS multi-frequency signals at a data sampling rate of 5 s. Three data processing schemes were set up: Scheme 1 used dual-frequency data, including BDS B1I/B2a, Galileo E1/E5a, and GPS L1/L2; Scheme 2 used triple-frequency data, including BDS B1I/B2a/B3I, Galileo E1/E5a/E5b, and GPS L1/L2/L5; Scheme 3 simultaneously used both dual-frequency and triple-frequency data. When performed RTK data processing, the software Net_Diff developed at the GNSS Analysis Center of Shanghai Astronomical Observatory was used [17].

For the combined BDS/Galileo/GPS data, a loosely coupled approach is employed, where each constellation utilizes its own base satellite for its respective DD observations. In addition, during parameter estimation, a kinematic forward Kalman filter is applied. For each satellite, the stochastic model is calculated using a weighting strategy that depends on the elevation angle.

When fixing the WL ambiguity and the raw ambiguity, the LAMBDA method is adopted. Furthermore, in order to improve the fixing rate, a partial ambiguity resolution (PAR) method based on the elevation angle strategy is utilized [22]. Additionally, to verify the reliability of AR, the ratio test is adopted with a threshold set to 3. The main data processing strategies are listed in Table 1.

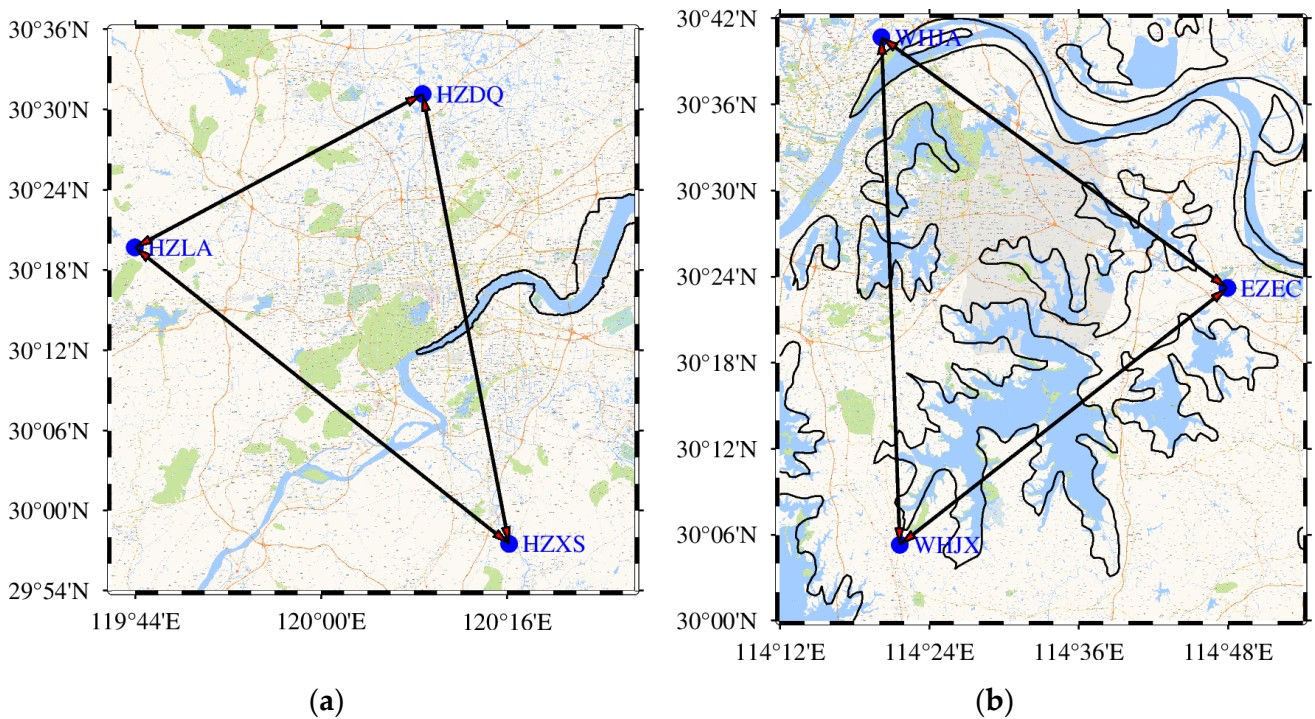

**Figure 1.** (**a**) Tree medium baselines in Hangzhou; (**b**) Tree medium baselines in Wuhan.

**Table 1.** Data processing strategies.

| Option | Setting |
|---|---|
| Elevation mask | 15° |
| SNR [1] | 30 dB |
| PDOP [2] | 20 |
| Code precision [3] | 0.3 m |
| Phase precision [3] | 0.003 m |
| Ionosphere delay for SPP [4] | Klobuchar model |
| Troposphere delay for SPP | Saastamoinen model with VMF1 mapping function |
| DD ionosphere delay | Estimated as random walk parameter |
| DD troposphere delay | Estimated as random walk parameter |
| Fix elevation mask [5] | 20° |
| AR mode | Continuous |

[1] Signal-to-noise ratio. [2] Position dilution of precision. [3] Zenith precision. [4] Single-point positioning. [5] AR is only performed for satellites with elevation angle greater than 20°.

## 4. DD Troposphere Delay and Ionosphere Delay Analysis

For medium baselines, the DD troposphere delay and ionosphere delay cannot be ignored. In order to analyze their influence on ambiguity resolution, it is necessary to study their magnitude. The satellite data, with ambiguity fixed, from six pairs of baselines are used for analyzing the DD troposphere delay and ionosphere delay residuals.

### 4.1. DD Troposhere Delay

Although the estimated DD troposphere delay in Equation (9) is in the zenith direction, it needs to be projected onto the station-satellite direction through a mapping function. In this section, we first conducted precise point-positioning ambiguity resolution (PPP-AR) at each station to derive precise slant troposphere delay [23,24]. Then, the DD slant troposphere delay of ambiguity-fixed satellites is constructed by using between-satellite single-difference and between-receiver single-difference. Figure 2 plots the DD slant troposphere delay for six baselines, with an elevation angle mask of 15°. It can be seen that the DD troposphere delay is negatively correlated with the elevation angle. When the elevation

angle is greater than 20°, the DD troposphere delay is generally less than 15 cm, which is less than a quarter of the wide-lane wavelength. Taking the baseline HZDQ-HZLA as an example, the root mean square error (RMSE) of DD troposphere delay for all ambiguity-fixed satellites is calculated and their mean value is 6.2 cm. As a result, it can be disregarded when solving the wide-lane ambiguity in Equation (8).

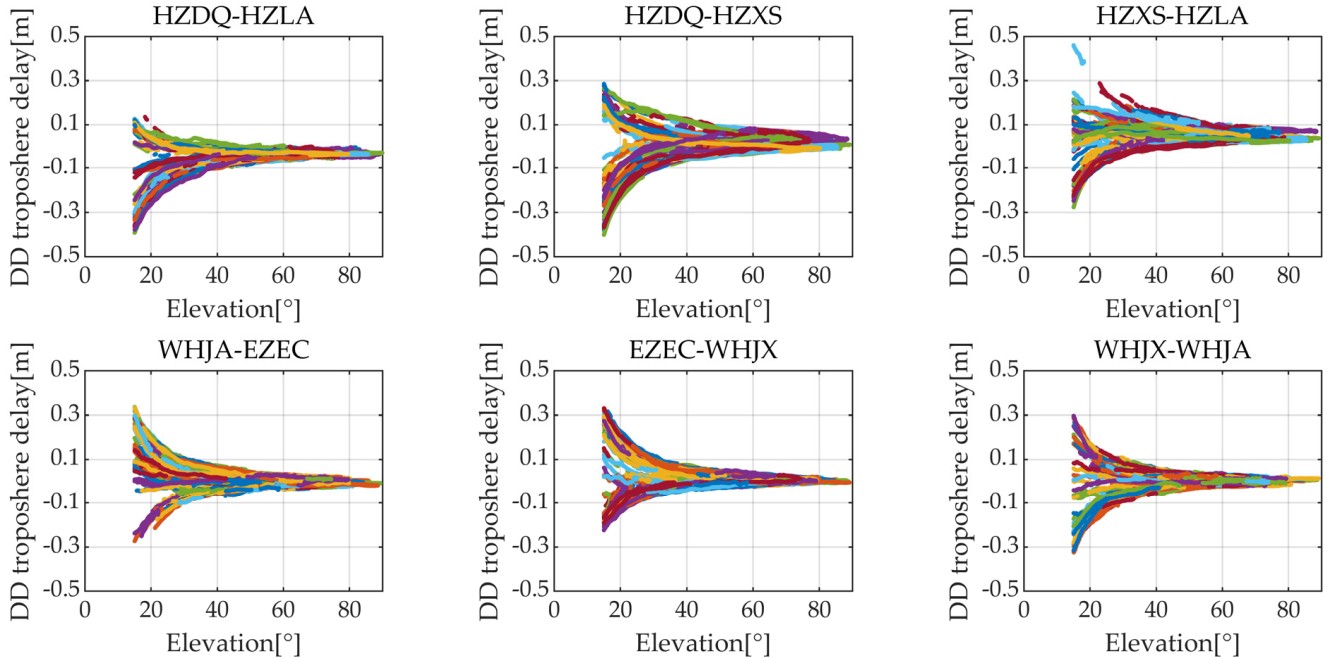

**Figure 2.** DD troposphere delay for HZDQ-HZLA, HZDQ-HZXS, HZXS-HZLA, WHJA-EZEC, EZEC-WHJX, and WHJX-WHJA. Each colored line represents a satellite.

*4.2. DD Ionosphere Delay*

As shown in Equation (9), the DD ionosphere delay is slant and satellite-dependent. In order to obtain precise DD slant ionosphere delay, after PPP-AR at each station, the geometry-free phase observation is used [20]. Then, the DD slant ionosphere delay for Frequency 1 (in meters) is also constructed by using between-satellite single-difference and between-receiver single-difference. Figure 3 plots the DD slant ionosphere delay for six baselines, with an elevation angle mask of 15°. It can be observed that, compared to DD troposphere delay series, DD ionosphere delay series variation is more severe. The DD ionosphere delay of some satellites can be significant, with values still exceeding 0.5 m when the elevation angle is greater than 40°. However, the majority of satellites exhibit a negative correlation between the DD ionosphere delay and the elevation angle.

Taking the baseline HZDQ-HZLA as an example, the RMSE of DD ionosphere delay for all ambiguity-fixed satellites is calculated, as shown in Figure 4. It can be seen that the DD ionosphere delay RMSE of all satellites is less than 0.3 m, with only four satellites having an RMSE greater than 0.2 m. The average RMSE for all satellites is 10.7 cm, which is smaller than the quarter-wavelength of the wide-lane wavelength; thus, it can be ignored when performing wide-lane ambiguity resolution.

Furthermore, considering the simultaneous impact of ionosphere delay and troposphere delay on the WL ambiguity resolution, we provide time series of float WL ambiguities based on Equation (8), using the HZDQ-HZLA as an example. Here, only ambiguities within ±5 cycles are plotted, as shown in Figure 5. It can be observed that the float WL ambiguities fluctuate mainly around integer cycles, demonstrating relatively good stability, indicating the feasibility of simultaneously neglecting DD ionosphere and troposphere delay residuals.

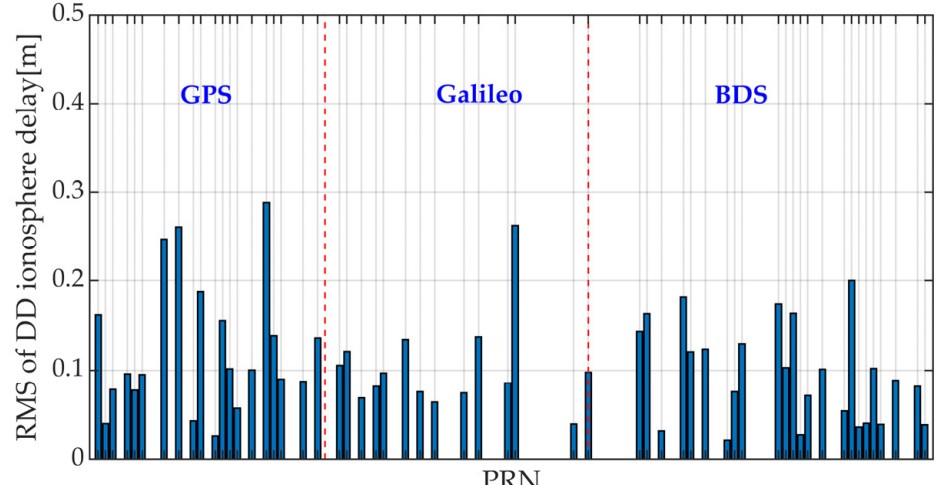

**Figure 3.** DD ionosphere delay of HZDQ-HZLA, HZDQ-HZXS, HZXS-HZLA, WHJA-EZEC, EZEC-WHJX, and WHJX-WHJA. Each colored line represents a satellite.

**Figure 4.** The RMSE of the DD ionospheric delay of all ambiguity-fixed satellites for the baseline HZDQ-HZLA.

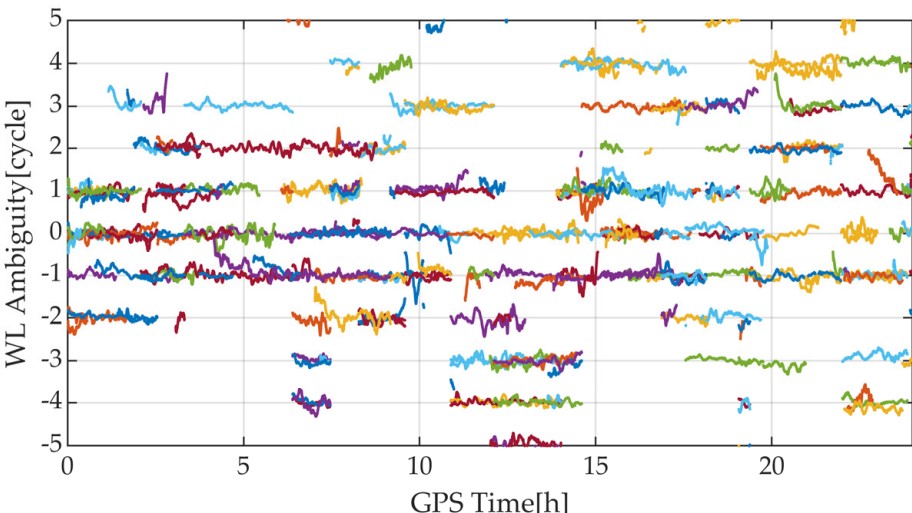

**Figure 5.** The time series of float WL ambiguities for the baseline HZDQ-HZLA. Each colored line represents a satellite.

## 5. Triple-Frequency Medium RTK Performance Analysis

### 5.1. Satellite Number and PDOP

Due to the baseline being a medium baseline, the number of available satellites and the PDOP can be analyzed using the HZDQ station as an example. Figure 6 plots the satellite number and PDOP of Scheme 2, i.e., only triple-frequency data are used and Scheme 3, i.e., both dual-frequency and triple-frequency data are used. It should be noted that the satellite number and PDOP for Scheme 1 are identical to those for Scheme 3; therefore, no analysis is conducted. It can be seen that, for Scheme 3, the number of satellites exceeding 90% of the epochs is greater than 17, while, for Scheme 2, it is 10. However, 10 satellites also fully meet the requirements for BDS/Galileo/GPS RTK positioning and provide faster computation speed. In addition, the average PDOP of both Schemes is less than 2, indicating a good geometric configuration [25].

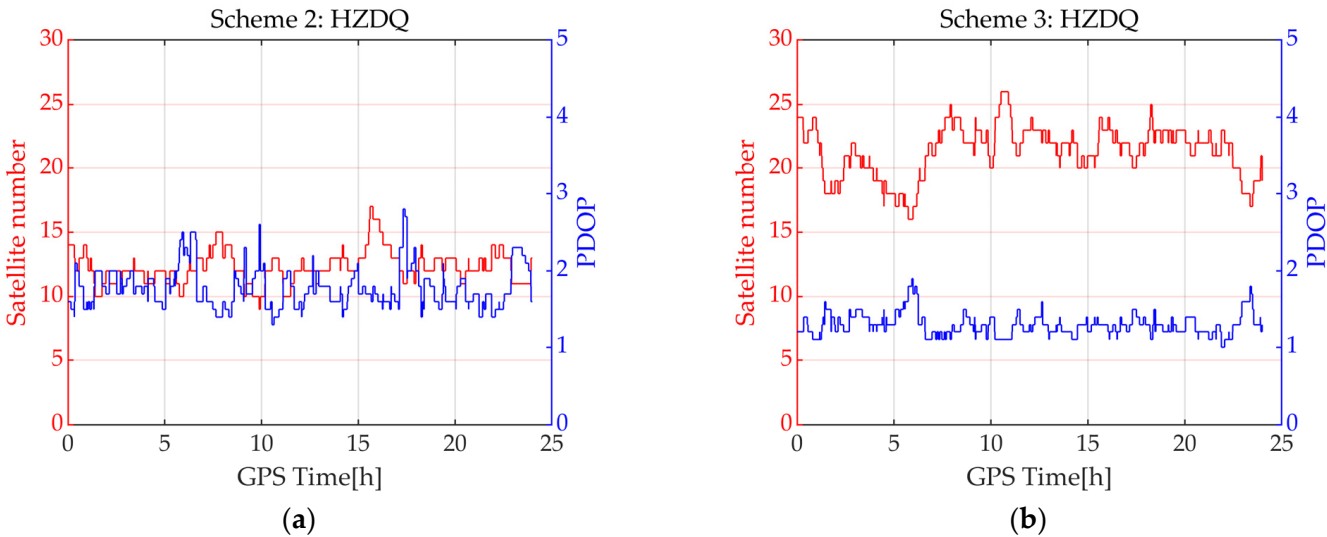

**Figure 6.** (**a**) The satellite number and PDOP of Scheme 2; (**b**) The satellite number and PDOP of Scheme 3.

### 5.2. Positioning Accuracy and Fixing Rate

Using the three schemes described in Section 3, we performed the RTK algorithm described in Section 2 for positioning. We calculated the positioning RMSE in both the

horizontal and vertical directions, as well as the fixing rate, which represents the proportion of correctly fixed solutions out of the total epochs, as shown Figure 7. A correctly fixed solution is defined as a solution with a ratio greater than 3.0 and a 3D positioning error less than 10 cm.

(**a**)

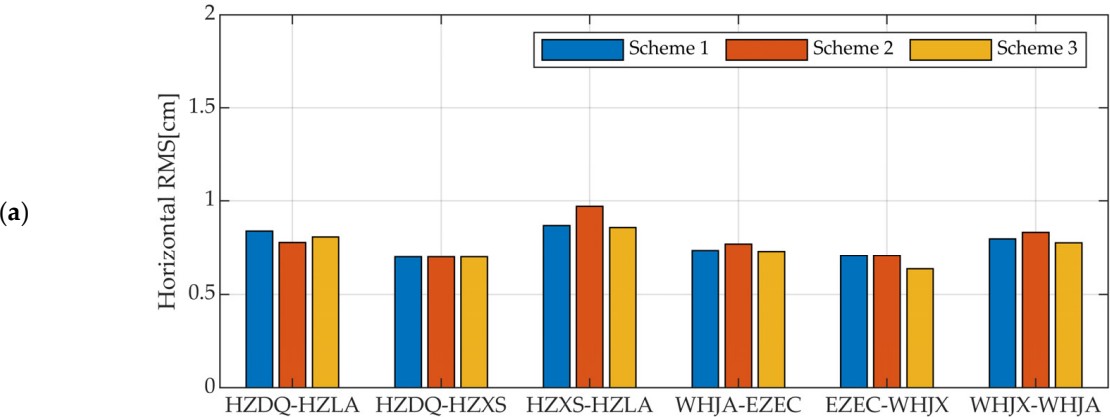

(**b**)

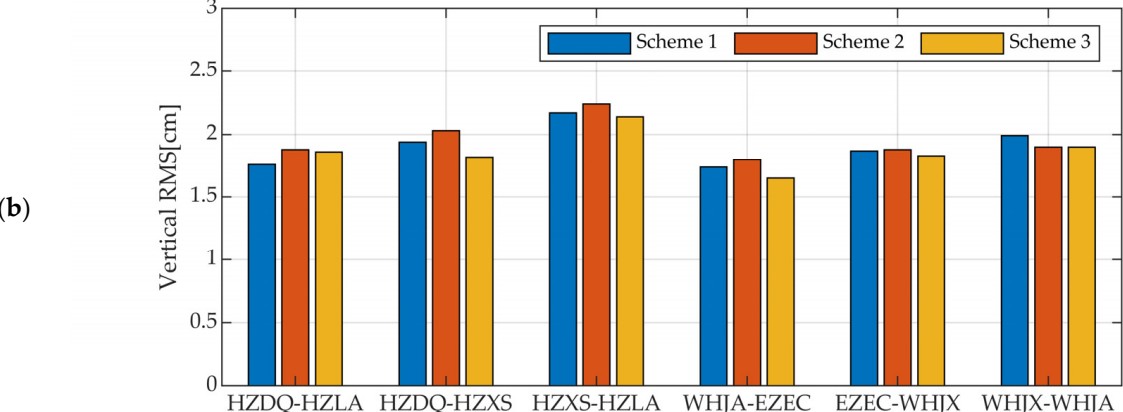

(**c**)

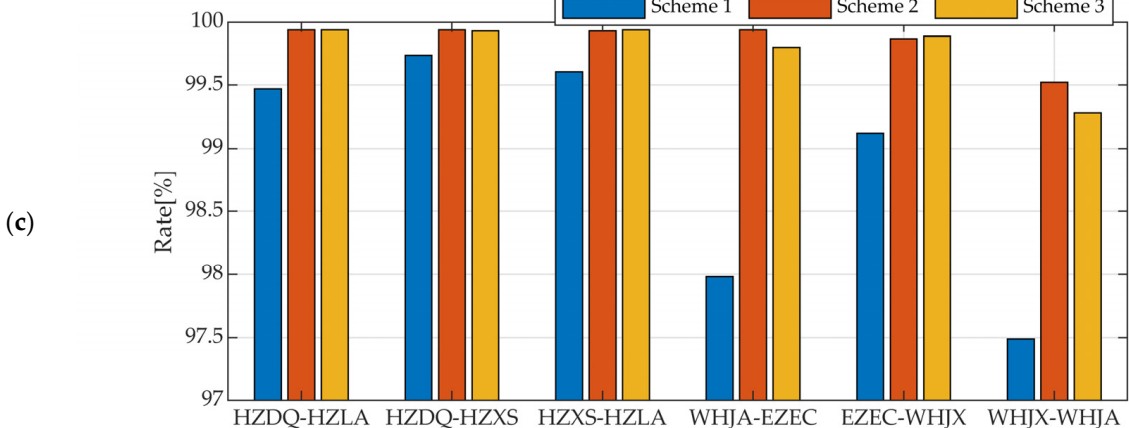

**Figure 7.** (**a**) Horizontal positioning RMSE of three schemes; (**b**) Vertical positioning RMSE of three schemes; (**c**) Fixing rate of three schemes.

Overall, the horizontal RMSE for the six baselines using the three positioning schemes is better than 1 cm, and the vertical RMSE is better than 2.5 cm. The horizontal and vertical RMSE difference for three schemes is less than 2 mm, which can be ignored for RTK

positioning. In terms of fixing rate, Scheme 2 and Scheme 3 are comparable and are both better than Scheme 1. Taking the baseline WHJX-WHJA as an example, Figure 8 shows the positioning error series using Scheme 1 and Scheme 2. It can be seen that, in the initial convergence stage, the triple-frequency scheme achieves instantaneous ambiguity fixing compared to the dual-frequency scheme, indicating that triple-frequency observations can accelerate convergence for medium baseline RTK. In addition, around 4 h and 8 h, the dual-frequency scheme has float solutions, while the triple-frequency-only scheme mostly has fixed solutions, indicating that the triple-frequency scheme is more likely to achieve ambiguity fixing.

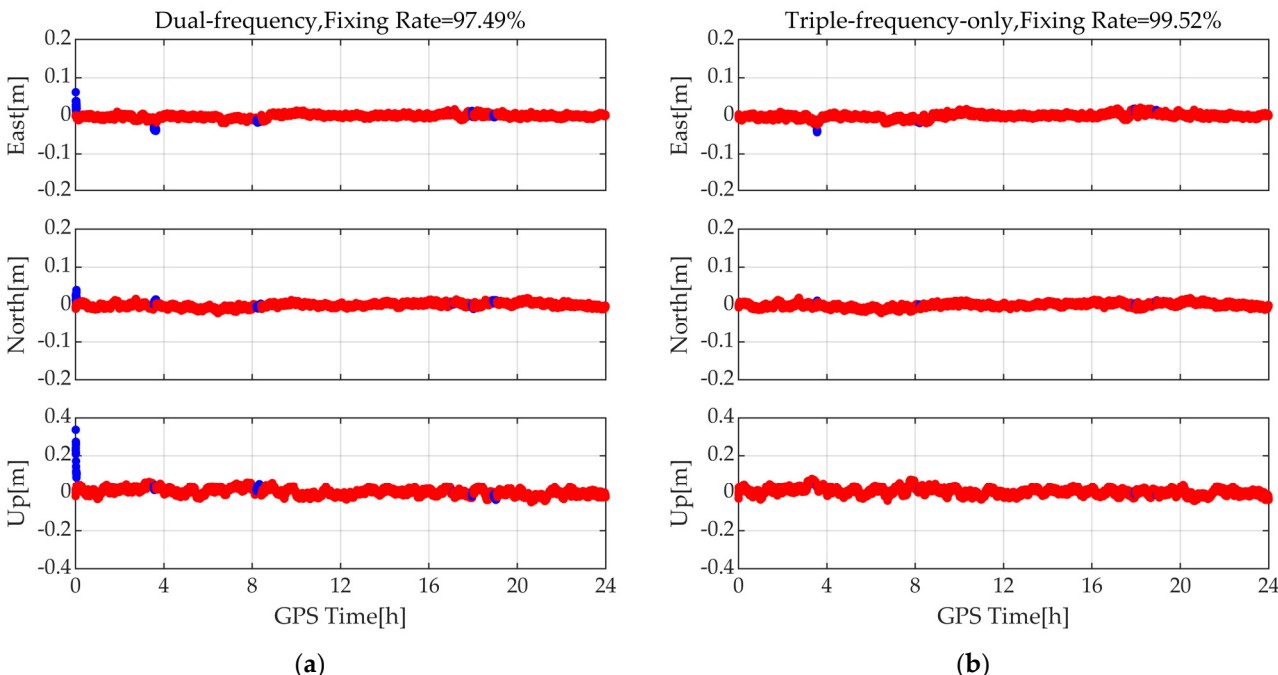

**Figure 8.** (**a**) The positioning error series of Scheme 1 for WHJX-WHJA; (**b**) The positioning error series of Scheme 2 for WHJX-WHJA. The blue point represents the float solution and the red point represents the fixed solution.

### 5.3. Convergence Performance

It is crucial for real-time users to have fast convergence, i.e., a 3D positioning error less than 10 cm, when satellite signals are obstructed. To evaluate such performance, we re-initialize the medium baseline RTK every 30 min. For each baseline, there are 48 sessions per day. The probability distribution of the convergence time for 48 sessions of 6 baselines is shown Figure 9.

It can be observed that, within 10 min, the probability of convergence for all six baselines using Scheme 2 and Scheme 3 exceeds 90%, whereas the baselines HZDQ-HZLA and WHJX-WHJA, which adopt Scheme 1, fall below 90%. Taking the baseline HZXS-HZLA, the convergence distribution for Scheme 2 and Scheme 3 is similar, with approximately 92% achieving convergence within 60 s, while, for Scheme 1, this value is only about 75%. For the baseline HZDQ-HZXS, the probability of achieving convergence within 60 s for Scheme 2 exceeds 95%, while the corresponding probability for Scheme 3 is approximately 82%, which is primarily attributed to the slower convergence of satellite ambiguities due to the absence of triple-frequency observations.

We calculated the probability of convergence time within 180 s for three baselines using the three different schemes, as shown in Table 2. The probability of convergence within 180 s is calculated to be approximately 82.6% for Scheme 1, 90.6% for Scheme 2, and 92.1% for Scheme 3. Compared to Scheme 1, both Scheme 2 and Scheme 3 show an approximate improvement of 8.0% and 9.5%, respectively, with both surpassing 90%.

Furthermore, compared to other baselines, the baseline WHJX-WHJA exhibits a lower probability of achieving convergence within 180 s. This is primarily attributed to the longer length of this baseline, which results in greater atmospheric spatial variations between 15:30 and 21:30.

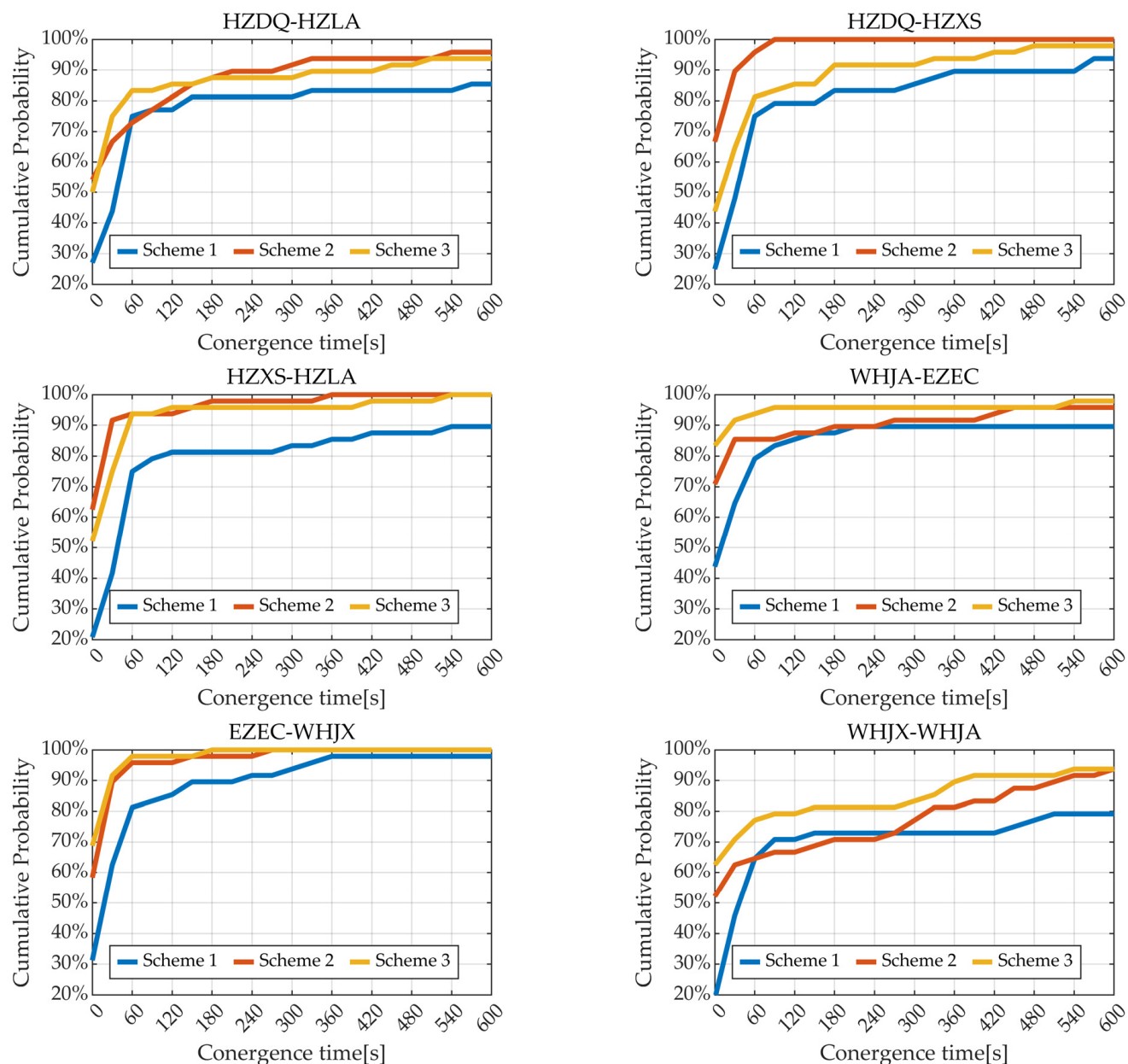

**Figure 9.** The probability distribution of the convergence time for 48 sessions of HZDQ-HZLA, HZDQ-HZLA, HZXS-HZLA, WHJA-EZEC, EZEC-WHJX, and WHJX-WHJA.

**Table 2.** The probability of convergence time within 180 s for three baselines.

| Baseline | Scheme 1 | Scheme 2 | Scheme 3 |
|---|---|---|---|
| HZDQ-HZLA | 81.3% | 87.5% | 87.5% |
| HZDQ-HZXS | 83.0% | 100.0% | 92.0% |
| HZXS-HZLA | 81.3% | 98.0% | 96.0% |
| WHJA-EZEC | 87.5% | 89.6% | 95.8% |
| EZEC-WHJX | 89.6% | 97.9% | 100.0% |
| WHJX-WHJA | 72.9% | 70.8% | 81.3% |

Using the baselines HZDQ-HZLA and WHJX-WHJA as examples, with resets occurring every 30 min, Figure 10 illustrates the positioning errors in the Up direction for three different schemes. It can be observed that Scheme 1, which only utilizes dual-frequency observations, has more float solutions and fixed solution ratios of only 84.27% and 80.00%, indicating slower convergence. However, Scheme 2 and Scheme 3, which incorporate triple-frequency observations, significantly reduce the number of float solutions, with a mean fixed solution ratio of approximately 91.7% and 92.7%. Additionally, it is worth noting that positioning convergence between 15 h and 20 h is relatively slow, primarily due to the high activity of the ionosphere in the day, as illustrated in Figure 11.

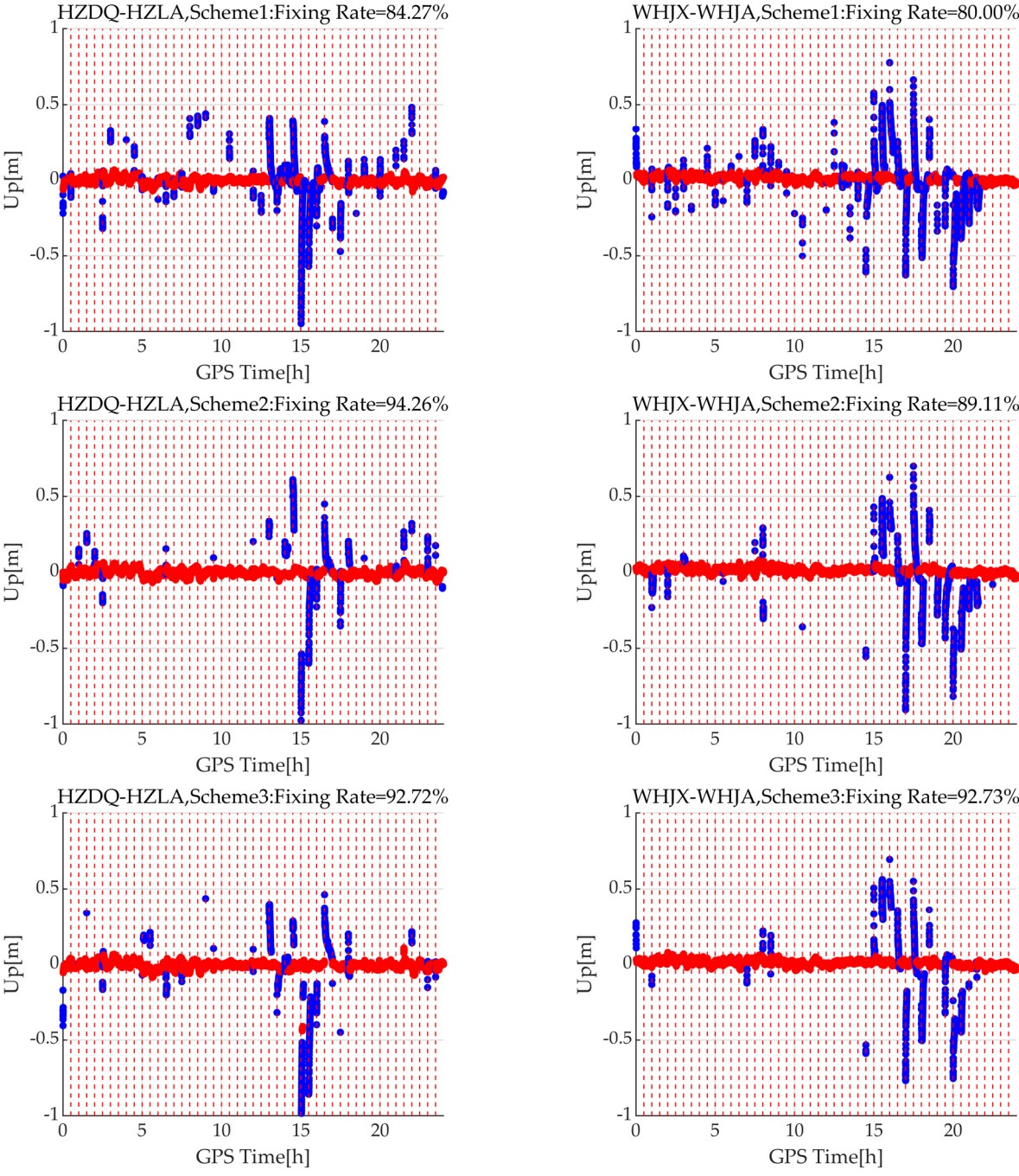

**Figure 10.** The positioning error in the Up direction for HZDQ-HZLA and WHJX-WHJA when using Scheme 1, Scheme 2, and Scheme 3. The red dots represent fixed solutions, while the blue dots represent float solutions.

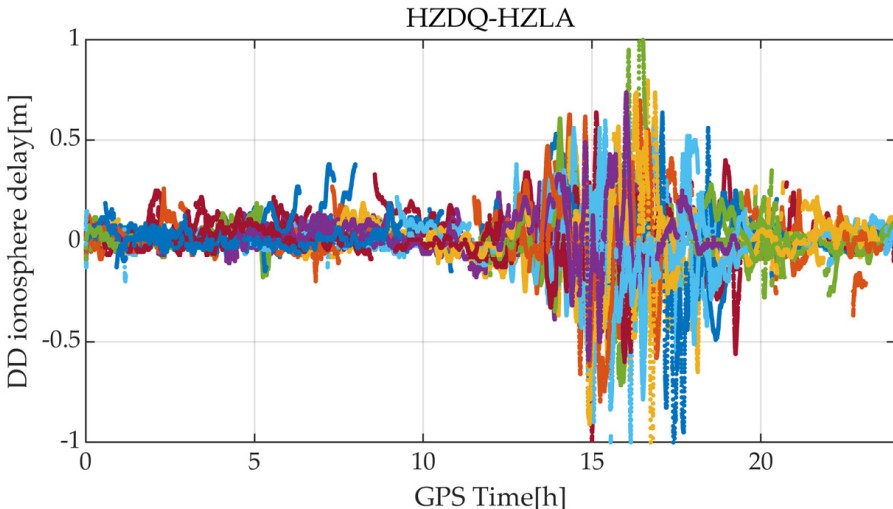

**Figure 11.** DD ionosphere delay time series of HZDQ-HZLA. Each colored line represents a satellite.

Furthermore, in the dual-frequency scheme (i.e., Scheme 1), BDS uses B1I/B2a, and, during the transitional phase, BDS2 satellites lack the B2a signal, rendering them unusable. Therefore, to increase the number of available BDS satellites, we conducted dual-frequency processing with B1I/B3I for the baseline HZDQ-HZLA and WHJX-WHJA, as depicted in Figure 12. The fixing rate of the baseline HZDQ-HZLA and WHJX-WHJA have increased by 1.3% and 0.9%, respectively, but they still significantly lag behind Scheme 2 and Scheme 3.

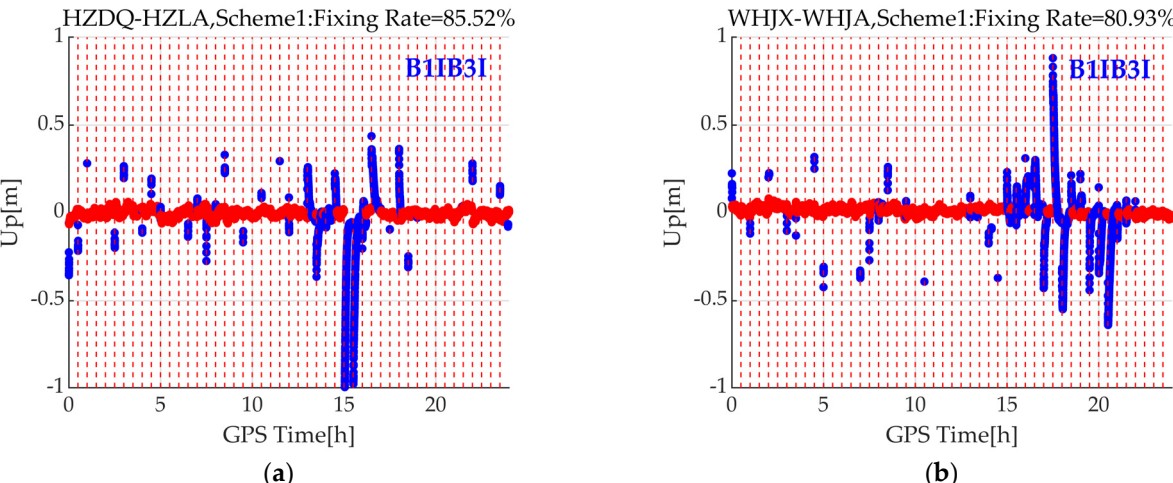

(**a**)   (**b**)

**Figure 12.** (**a**) The positioning error in the Up direction for HZDQ-HZLA, using dual-frequency BDS B1I/B3I and (**b**) The positioning error in the Up direction for WHJX-WHJA, using dual-frequency BDS B1I/B3I. The red dots represent fixed solutions, while the blue dots represent float solutions.

### 5.4. Computation Cost Time

For high-frequency terminals, such as those operating at 10Hz, the computational time of the RTK algorithm is an essential consideration. Excessive computational power consumption will lead to an increase in hardware costs. Considering the advantage of accelerated convergence through triple-frequency observations, this section compares the computational time of Scheme 2 and Scheme 3. Based on the Intel i7-10700 processor with a clock frequency of 3.8 GHz, the processing time of 48 sessions for the 6 baselines is shown in Figure 13.

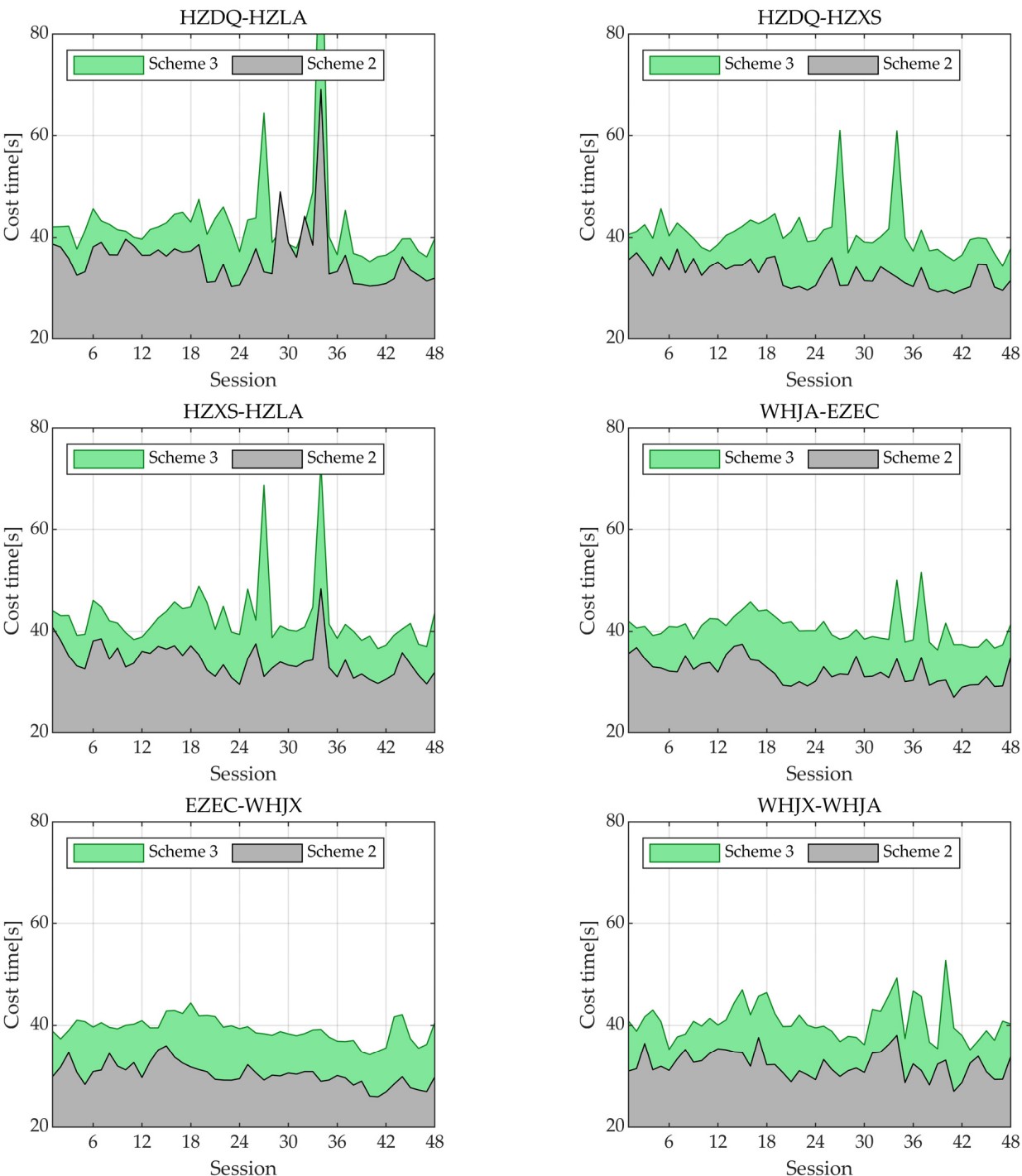

**Figure 13.** The processing time of 48 sessions for 6 baselines when using Scheme 2 and Scheme 3.

Taking the baseline HZDQ-HZLA as an example, except for sessions 29, 34, and 36 (which exceed 40 s), all other sessions take less than 40 s for Scheme 2. In contrast, for Scheme 3, most sessions exceed 40 s, with session 34 being the longest at over 80 s. For other baselines, most sessions in Scheme 2 are also completed in under 40 s, whereas, in Scheme 3, the majority of sessions require more than 40 s. Table 3 provides statistics on the average computation time for the 48 sessions of the 6 baselines. Compared to Scheme 3, Scheme 2 has an average time consumption reduction of 8.26 s, or approximately 20%. The scheme that utilizes triple-frequency-only observations has a lower computation time, being more beneficial for high-frequency GNSS terminals.

**Table 3.** The average duration of the 48 sessions for the 6 baselines.

| Baseline | Scheme 2 | Scheme 3 |
|---|---|---|
| HZDQ-HZLA | 36.00 s | 42.85 s |
| HZDQ-HZXS | 32.74 s | 40.91 s |
| HZXS-HZLA | 34.14 s | 42.77 s |
| WHJA-EZEC | 32.07 s | 40.61 s |
| EZEC-WHJX | 30.33 s | 39.17 s |
| WHJX-WHJA | 32.23 s | 40.79 s |

## 6. Discussion

By constructing extra-wide-lane observations to expedite ambiguity convergence, triple-frequency RTK has gained extensive research attention in recent years. However, many studies are based on mixed-frequency models (i.e., using both dual-frequency and three-frequency observations), mainly due to the limited availability of triple-frequency satellites during the research. With the provision of global services by BDS3 and Galileo, the number of available triple-frequency satellites has significantly increased. Therefore, in this paper, we conducted RTK with triple-frequency-only observations for six medium baselines. Experimental results show that triple-frequency-only RTK has little difference in positioning accuracy, fixing rate, and convergence time compared to the mixed-frequency model. However, the computation time of triple-frequency-only method is significantly shorter than the mixed-frequency model, which is beneficial for high-frequency on-embedded terminals.

Compared to the RTK results in the high-latitude Tokyo area in [17], this study uses medium baselines at lower latitudes, where the double-difference ionosphere and troposphere delay residuals are more pronounced. However, their impact on wide-lane ambiguity resolution can still be ignored. Furthermore, for the 60 km baseline, this study's convergence performance using the GPS/Galileo/BDS three-system combination is significantly better than [17], which uses GPS/Galileo, indicating that the BDS can enhance multi-system RTK performance.

Although there are the encouraging findings revealed by this paper, there are also limitations. First, the baselines we used are still situated at mid-latitudes, and the performance of the RTK method proposed in this paper in low-latitude regions remains to be evaluated. This article employs static station simulation for RTK positioning, without considering the performance of medium baseline RTK positioning in kinematic scenarios for vehicles. Moreover, due to the high quality of the observations, there is currently a great emphasis on the convergence time of RTK, with less attention given to its fixed reliability. In future work, further experiments and analysis will be conducted to address these issues.

## 7. Conclusions

In this paper, the ambiguity resolution method of triple-frequency medium baseline RTK are presented in detail, including the geometry-free EWL AR and the fast geometry-based WL AR. The DD troposphere delay and ionosphere delay residuals are presented and their impact on AR is analyzed. Furthermore, based on six medium baselines, the RTK positioning performance using dual-frequency data, triple-frequency-only data, and both dual-frequency and triple-frequency data is evaluated in terms of the number of satellites and PDOP, positioning accuracy, fixing rate, convergence performance, and computation cost time. The experimental analysis shows the following:

- For medium baselines of 45–66 km at latitude $30°$, the RMSE of DD slant troposphere delay is about 6.2 cm, and the RMSE of DD slant ionosphere delay is about 10.7 cm. They can be neglected for geometry-based WL ambiguity resolution, but cannot be neglected when fixing the raw ambiguity;

- In the Yangtze River Delta region of China, when performing BDS/Galileo/GPS triple-system positioning, 90% of the time includes more than 10 available satellites with 3 frequencies (BDS B1I/B2a/B3I, GPS L1/L2/L5, Galileo E1/E5a/E5b). The average

PDOP value of triple-frequency-only case during the entire time period is less than 2.0, indicating a good geometric configuration;

- Compared to dual-frequency RTK, the improvement in accuracy after convergence is not obvious for triple-frequency RTK, but the convergence speed is improved. Furthermore, compared to dual-frequency RTK, the probability of completing convergence within 180s is increased by about 8.0% for triple-frequency-only RTK;
- Compared to the scheme of using both dual-frequency and triple-frequency data simultaneously, the computation cost time of the scheme using triple-frequency-only data is reduced by 8.26 s, improving by approximately 20%.

**Author Contributions:** Conceptualization, X.D. and C.G.; methodology, X.Y.; software, Y.Z.; validation, J.W. and Y.L.; writing—original draft preparation, X.D. and X.Y.; writing—review and editing, C.G.; visualization, X.Y. All authors have read and agreed to the published version of the manuscript.

**Funding:** This research was funded by the Innovation project of China Railway First Group (2022A-031), the Research and Development Fund Project of Zhejiang A&F University (2021LFR030), and the National Natural Science Foundation of China (42074014).

**Data Availability Statement:** All data can be made available by contacting the corresponding author.

**Conflicts of Interest:** The authors declare no conflict of interest.

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
