# Peer review of "Improved Medium Baseline RTK Positioning Performance Based on BDS/Galileo/GPS Triple-Frequency-Only Observations"

_remotesensing, doi:10.3390/rs15215198_

Round 1
Reviewer 1 Report (Previous Reviewer 1)
Comments and Suggestions for Authors
The re-submitted manuscript does not address the problem that I raised in my previous reviewing. The work described in this manuscript actually doe not make much sense. The key issue comes from Eq.(9). In the equation, the carrier phases of the first frequency and the second frequency are used twice in observation domain. The equation is rank deficiency if the WL ambiguities are not solved beforehand. It is not recommended to do WL ambiguity fixing beforehand through a geometry-free and geometry-based estimation. The procedure just makes the procedure complex without gaining in ambiguity-fixing and positioning. The methodology in Eq. (6-9) is not used in GNSS community for GNSS relative positioning, therefore, comparing with it makes no sense.
The work can be very simple if the authors just start from Eq. (1). They can either form wide-lane or extra wide-lane ambiguities in design matrix domain (i.e. explicit decorrelation), or rely on LAMBDA method to decorrelate the ambiguities (i.e. implicit decorrelation). Then the authors could compare solutions produced from 1) dual-frequency measurements, 2) both dual-frequency and triple-frequency measurements and 3) triple-frequency only from the point view of positioning accuracy and convergence.
In addition, it would make sense to use B3I as the second frequency instead of B2a for BDS, in this case, more dual-frequency BDS satellites can be used in Scheme 1.
To the manuscript more scientific soundness, I would suggest the authors consider to make experiments in ionosphere active period to assess the performance of above three schemes. Otherwise, the manuscript is lack of academic sense.
Comments on the Quality of English LanguageThe English presentation can be better if it is smoothed by an professional agent.
Author Response
Please see the attachment.

Reviewer 2 Report (Previous Reviewer 2)
Comments and Suggestions for Authors
The work “Improved Medium Baseline RTK Positioning Performance Based on BDS/Galileo/GPS Triple-Frequency-Only Observations” demonstrates a case scenario's results for triple-frequency RTK; thus, its novelty is limited. The work could be accepted after providing the following clarifications/improvements.
- Since this a resubmitted work, I still do not see the clarification of differences between dual and triple frequency RTK, particularly concerning ambiguity resolution (in the former case, we cannot use EWL combination)
- In section 2, it should be informative to show an example of time series of combination variation used for EWL computation (equation 6). Just a single data arc would be helpful. The same situation could be done for WL as well. Such figures would support your further statements about the possibility of neglecting DD tropo and DD iono residuals. Please note that you analyze both atmospheric factors separately, but they co-occur.
- Page 6, table 1, The authors still do not explain why 0.3 m precision was applied for code data. Was it verified in any way?
- In the same table, was SPP computed with a single frequency? (if you use the Klobuchar model). Typically, an iono-free combination of code data is used for this purpose. I would expect this is more stable than Klobuchar model for a single frequency.
- In lines 294-302, the authors provide some statistics for 10-minute and 60-second delays, then in table 2 for 180 seconds. Is there a lot of such numbers needed? Much more interesting would be a discussion of why the results differ between particular baselines (see vector WHJX-WHJA).
- In lines 320-321, you write, “it is worth noting that positioning convergence between 15h~20h is relatively slow, primarily due to the high activity of the ionosphere” but in conclusion, it is stated the ionosphere was quiet. This is not clear at all. If it was an ionosphere, please provide any indicator of this activity. Moreover, according to your algorithm description, the ionosphere should not influence AR.
- Lines 368-372, “It should be pointed out that in this study of DD atmospheric residuals, the Kp index ranges from 0.3o to 2.3o, indicating a relatively quiet ionosphere and the absence of analysis for active ionospheric disturbances. It is necessary to analyze whether the DD ionosphere delay can be disregarded for WL ambiguity resolution during the active period of the ionosphere.” The information on Kp was the first time given in conclusions. This seems to be without sense. The ionospheric conditions are not clear, see the comment above.
Author Response
Please see the attachment.

Reviewer 3 Report (Previous Reviewer 3)
Comments and Suggestions for Authors
Dear Authors,
after your revision of the manuscript I believe it is ready for publication.
Author Response
Thank you for your suggestion; the article has been revised.
Round 2
Reviewer 1 Report (Previous Reviewer 1)
Comments and Suggestions for Authors
no further comments
Comments on the Quality of English Languageno further comments
Reviewer 2 Report (Previous Reviewer 2)
Comments and Suggestions for Authors
The Authors have addressed my concerns, and I think the work can be published.
This manuscript is a resubmission of an earlier submission. The following is a list of the peer review reports and author responses from that submission.
Round 1
Reviewer 1 Report
Comments and Suggestions for Authors
The entire work described in this manuscript actually does not make much sense. The key issue comes from Eq.(9). In this equation, the carrier phases of the first frequency and the second frequency are used twice in observation domain. The equation is rank deficiency if the WL ambiguities are not solved beforehand. It is not recommended to do so, even the WL ambiguities are solved beforehand through a geometry-free and geometry-based estimation. It just makes the procedure complex without gaining in ambiguity-fixing and positioning, which is demonstrated in later sections of the manuscript. The methodology in Eq. (6-9) is not used in GNSS community for GNSS relative positioning, therefore, comparing with it makes no sense.
In addition, the troposphere parameter would be satellite independent, because the mapping function is applied, therefore one troposphere parameter is enough.
The authors would compare the solutions using satellites having only triple frequency measurements and the solutions using satellites having both triple and dual frequency measurements. Then the authors will probably come up with different conclusions.
Comments on the Quality of English LanguageThe authors are encouraged to use the written language, not the oral ones, such as "got", "very", ...
Reviewer 2 Report
Comments and Suggestions for Authors
The work “Improved Medium Baseline RTK Positioning Performance Based on BDS/Galileo/GPS Triple-Frequency-Only Observations” provides another test of triple-frequency RTK, and thus its novelty is limited. Nevertheless, I believe the work could be accepted after addressing the issues below.
- The work compares dual- and triple-frequency RTK, but provided algorithm (section 2) only refers to the latter approach. Certainly, dual-frequency RTK is well known, but the differences between both RTK algorithms should be clarified in section 2.
- Page 5, line 163, Why such values (1.5 m and 0.2 m) were applied? According to Figure 3, the variance for DD ionospheric delays is expected to be lower. A similar situation is for DD troposphere.
- Page 6, table 1, The septentrio receivers have low measurement noise. I think for code data it is lower than 0,3 m, so why such values?
- Pages 5 and 6, on page 5, it is stated that a priori DD ionospheric delay is set to 0, but according to table 6, it is taken from the Klobuchar model. Can such a simple model be useful?
- Additional PPP-AR tests support the results of DD tropospheric delay given in Figure 2. Why did you not use the original RTK tests?
- Page 7, what does it mean “does not a projection transformation”?
- Section 5.1, If I am right, the number of satellites is the same for schemes 1 and 3. If yes, please indicate it.
- Page 10, I would not say that change of fixing rate from 99,47% to 99,94% is a significant (Figure 7). The entire difference is related to faster convergence for schemes 2 and 3. Thus, there are better ways to show this effect than 24-hour analysis. It is much clearer explained in section 5.3.
- The results given in Figure 8b are partly unexpected for me. Why is the outcome for scheme 2 (with a relatively low number of observed satellites) characterized by much faster convergence? Please discuss it.
- The results in Figure 9 indicate a lower convergence time, around 15 h. What can be a reason for this effect
- In my opinion, section 5.4 should be removed.
Comments on the Quality of English Language
Some sentences are difficult to follow. Please read carefully the entire text
Reviewer 3 Report
Comments and Suggestions for Authors
Dear authors,
thanks for writing this nice article. Even though no completely new development was shown in your work, I believe that the content of your paper is useful for people dealing with multi-frequency multi-systen RTK.
I have the following comments that you could implement to improve the article:
- The Captions of Figures 2,3,5,7,9 and need to be improved. Please explain the colors of the different data either in the caption or the plots themselves
- It is not clear to me what unit is shown in Figure 3 and 4. Is it the double-difference ionosphere effect on the L1 frequency? If yes, then you should specify this somewhere. Please also consider that the influence on the widelane combination will be slightly bigger.
- Please correct the Figure 10 caption. You are not showing a 3D positioning error time series there.
- Regarding your discussion I am missing some information on the false-fix rate. You do show that occasionally you see iono residuals on the order of a wavelength (Fig.3) so that I would expect to see some false fixes in your positioning solutions. Is this true? If yes, with which rate do they occur? Did you try to optimize the tradeoff between convergence time and false fix probability? If not, could you include this in your discussion?
I am looking forward to see the final article being published.
Reviewer 4 Report
Comments and Suggestions for Authors
This manuscript assessed the RTK performance using multi-GNSS data. However, the experiment seems too simple, from my personal point, more experiment data is required. Meanwhile, the title is “improved ….”, but I can not found any innovation of this manuscript. Therefore, I personally can not recommend this manuscript for publication. The following are the details which could be checked by the editor.
1. The title of this article is "Improved Medium Baseline RTK Positioning Performance Based on BDS/Galileo/GPS Triple-Frequency-Only Observations," I can not find what is improved.
2. the experiment is not sufficient to support the conclusion, where only three sets of static data from the same area within a day are used.
3. Line 180, the article adopts a loosely coupled approach. Why wasn't a tightly coupled approach considered? For medium baseline RTK, what are the differences between results of these two methods?
4. Section 4.1 and 4.2, which satellite are used to compute the DD tropospheric delay and ionospheric slant delay, respectively.
5. Line 227, how to discern a negative correlation between ionospheric slant delay and elevation angle from the Figure 3?